# SLIBO-Net: Floorplan Reconstruction via Slicing Box Representation with Local Geometry Regularization

**Jheng-Wei Su**[1], **Kuei-Yu Tung**[1], **Chi-Han Peng**[2], **Peter Wonka**[3], **Hung-Kuo Chu**[1,†]

[1]National Tsing Hua University [2]National Yang Ming Chiao Tung University
[3]King Abdullah University of Science and Technology
[†]Correspondence: `hkchu@cs.nthu.edu.tw`

## Abstract

This paper focuses on improving the reconstruction of 2D floorplans from unstructured 3D point clouds. We identify opportunities for enhancement over the existing methods in three main areas: semantic quality, efficient representation, and local geometric details. To address these, we presents SLIBO-Net, an innovative approach to reconstructing 2D floorplans from unstructured 3D point clouds. We propose a novel transformer-based architecture that employs an efficient floorplan representation, providing improved room shape supervision and allowing for manageable token numbers. By incorporating geometric priors as a regularization mechanism and post-processing step, we enhance the capture of local geometric details. We also propose a scale-independent evaluation metric, correcting the discrepancy in error treatment between varying floorplan sizes. Our approach notably achieves a new state-of-the-art on the Structured3D dataset. The resultant floorplans exhibit enhanced semantic plausibility, substantially improving the overall quality and realism of the reconstructions. Our code and dataset are available online[1].

## 1   Introduction

Reconstructing 2D floorplans from unstructured 3D point clouds has received significant attention in recent years as it facilitates various applications in scene understanding, robotics, VR/AR, etc. Prior works [24, 8, 9, 22, 14] typically employ a series of operations to address the problem at hand. The raw 3D point cloud is initially projected along the gravity axis (or z-axis) to generate a 2D density map. This density map is then fed into a deep neural network to either extract room masks [8, 22] or regress room corners [9, 14], followed by an optimization process to extract room polygons. In contrast to the two-stage framework, Yue et al. [24] introduced the RoomFormer model, an end-to-end trainable, transformer-type architecture that directly predicts room polygons using two-level queries. Although deep learning approaches have exhibited impressive performance in floorplan reconstruction, we have identified the following opportunities for improvement.

**Semantic quality.**   The plausibility or semantic quality of reconstructed floorplans heavily relies on the accuracy of estimated room corners and masks. Insufficient supervision of the room shape or fragile room representations can result in artificial room shapes, as demonstrated in Figure 1(a), that are not plausible.

**Efficient representation.**   One main challenge is designing a representation that requires only a reasonable number of tokens in the transformer architecture and can handle a variable number of rooms, each with a variable number of corners. For example, the naive approach of predicting the layout room by room, corner by corner does not require too many tokens but is too difficult to

---

[1]`https://ericsujw.github.io/SLIBO-Net/`

37th Conference on Neural Information Processing Systems (NeurIPS 2023).

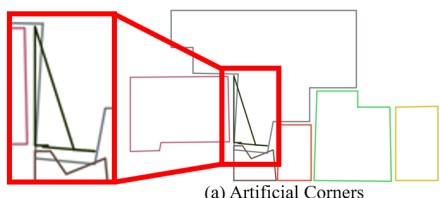 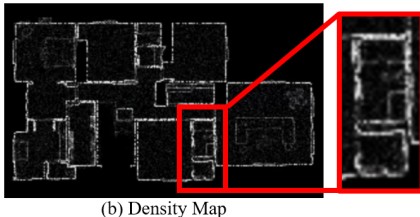 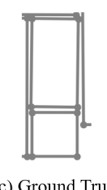

| (a) Artificial Corners | (b) Density Map | (c) Ground Truth |

Figure 1: **Limitations for current methods.** (a) The lack of supervision on the overall room shape can lead to the generation of artificial corners. (b) The density map of a large floorplan may have insufficient resolution to effectively capture local geometric details, such as wall thickness, which are clearly evident in the (c) ground truth layout.

train. To address this challenge, we employ a novel floorplan representation that slices the floorplan into a set of boxes by scanning it in a top-down manner (Figure 3). This representation offers two key advantages. Firstly, it yields a manageable number of slicing boxes, making it suitable for the transformer architecture as each slicing box can be represented by a query token. Secondly, these slicing boxes provide improved supervision of the room shape by utilizing Intersection over Union (IoU) loss [15]. Furthermore, this representation naturally leads to higher semantic quality, enhancing the overall accuracy and realism of the reconstructed floorplans.

**Local details.** We observed that deep learning approaches excel in predicting the overall room structure but struggle to capture local geometric details such as wall thickness due to image resolution limitations (Figure 1(b)). We incorporate prior knowledge about geometric details derived from the input 3D point cloud to address this limitation. This knowledge is encoded as cell complexes, which serve multiple purposes. It acts as a regularization mechanism during the training phase and is also utilized as a post-processing step to enhance the accuracy of the reconstructed floorplans.

**Evaluation.** The metrics employed in previous studies are scale-dependent, resulting in differential treatment of errors between large and small floorplans. To illustrate this, consider starting with a floorplan of a specific size. Adding additional rooms to the floorplan alters its error measurement, leading to a more lenient evaluation of incorrect corner predictions. This behavior is counter-intuitive and primarily masks a drawback of neural networks in scaling to larger inputs. Hence, we advocate for adopting a scale-independent metric that mitigates this issue.

In summary, we make the following contributions: 1) We propose a novel transformer architecture for floorplan reconstruction that only requires a manageable number of tokens. 2) We propose integrating geometric priors as regularization during training and as a post-process to improve local geometric details. 3) We advocate the use of a new metric that is scale independent and that better measures if local details are correctly reconstructed. 4) We produce more semantically meaningful floorplans that do not suffer from implausible geometric details.

## 2 Related Work

**Optimization-based methods.** Early work mostly assumed the task to be based on finding 3D planes from the point clouds using traditional techniques such as Hough transform [16, 2], plane sweeping/plane fitting [4, 23, 17, 12, 18, 21], and leveraging histograms of point cloud positions [7]. Higher-level structured objects such as rooms could then be assembled from the recovered planes. For larger buildings with many rooms, the optimization became increasingly difficult to solve. Researchers then developed graph-based data structures [10, 5, 11, 8, 22] to help organizing the reconstructed sub-models (e.g., corners, walls, objects, and rooms) and guide the optimization to follow certain semantic, topological constraints. In particular, Ikehata et al. [11] used hierarchical tree graphs of reconstructed sub-models, such as objects, walls, and rooms, to guide the 3D reconstructions. Bassier and Vergauwen [3] extracted highly-detailed BIM-compatible objects from point clouds by plane fitting and RANSAC. In the same spirit of extracting more semantic and realistic 3D models, Ochmann et al. [19] proposed methods to extract *parametric* 3D building models from input point clouds. A key advantage was that their reconstructed floorplans are volumetric - i.e., having wall thickness. Our results shared the same idea. Finally, our cell complex-based constraints was inspired

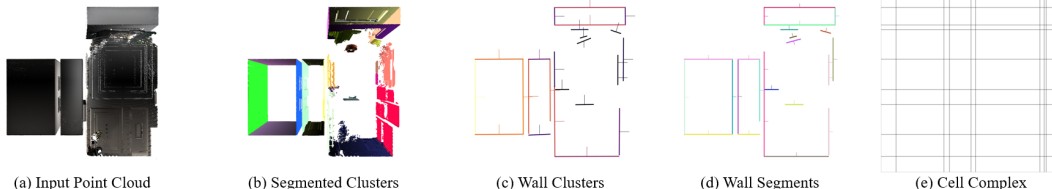

| (a) Input Point Cloud | (b) Segmented Clusters | (c) Wall Clusters | (d) Wall Segments | (e) Cell Complex |

Figure 2: **Encoding geometric priors into the cell complex.** (a) Input 3D point cloud. (b) Segmentation based on normals and proximity. (c) Plane fitting and grouping. Aerial view of wall clusters with orientations. (d) Wall segment formation by collinear line grouping. (e) Cell complex from parametric lines. Non-axis aligned lines removed.

by [13], in which they proposed an optimization-based approach to register multiple 3D scans of multi-story, multi-room buildings.

**Deep learning methods.** More recent methods began to leverage neural networks to solve the floorplan reconstruction problems. Earlier methods were hybrid methods that utilized both optimization and neural networks. Floor-SP [8] assembled room segments detected by Mask R-CNN [1] to form full floorplans by sequentially solving shortest path problems. MonteFloor [22] later solved the room assembly problem by Monte-Carlo Tree-Search (MCTS). FloorNet [14] took a point cloud and RGB images as inputs, utilized a triple-branch neural network to generate pixel-wise predictions on floorplan geometry and semantics, and then used an integer programming to generate the final floorplans in vector-graphics format.

Later, fully neural network-based methods (which are end-to-end trainable) were being developed and delivered state-of-the-art performances. HEAT [9] was a method to reconstruct planar graphs with underlying geometric structures from 2D raster image inputs. It had two main applications - building segmentation in aerial images and floorplan reconstruction from 2D density maps distilled from point clouds. It utilized an end-to-end trainable edge classification neural network for the task. Recently, RoomFormer [24] proposed a single-stage, end-to-end trainable neural network to directly predict floorplans from a given point cloud. Their main innovation was to encode a floorplan as variable-size set of polygons, which are variable-length sequences of ordered vertices. This design enabled efficient handling by transformers. Our method is an improvement upon it.

# 3 Methodology

## 3.1 Pre-processing

In the data preprocessing stage, we adopt a similar process as in previous works [8, 9, 14, 22] to project the raw 3D point cloud along the gravity axis (or z-axis), resulting in a 2D density map in the aerial view (or x-y plane). It is important to note that while this density map provides adequate visual information for describing the overall structure (e.g., rooms and walls) of the floorplan, it faces challenges in capturing geometric details like wall thickness due to the limitation of image resolutions.

**Encoding geometric priors.** We extract wall segments from 3D point cloud data as geometric priors for floorplan reconstruction. We project these wall segments as parametric lines onto the x-y plane, forming a "cell complex" representation. Figure 2 shows the steps of computing a cell complex from a 3D point cloud. First, we segment the point cloud by normals and proximity (Figure 2(b)). Next, we fit planes for each cluster and select those orthogonal to gravity as wall clusters (Figure 2(c)). After that, we project points onto planes and group collinear line segments as wall segments (Figure 2(d)). Finally, we convert wall segments into parametric lines and remove non-axis aligned lines to get the cell complex (Figure 2(e)). After obtaining the cell complex representation, we follow the strategy described in Section 3.4, where we adjust our ground truth by rounding it to the cell complex representation for training.

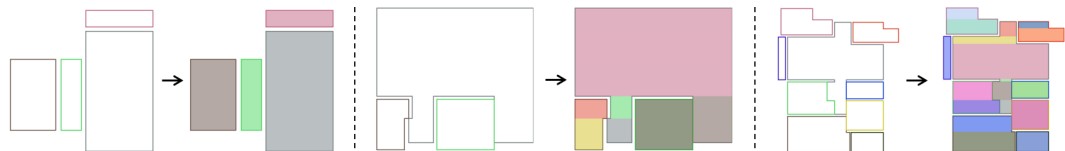

Figure 3: **Floorplan representation.** Illustrative examples highlighting floorplans (left) and their respective slicing boxes representation (right).

## 3.2 Floorplan representation

One key challenge in our work was designing a suitable data structure for floorplans. The representation should work with the transformer architecture and only require a reasonable number of tokens. One possible representation would be to represent each grid cell of the cell complex as a query token, but this would require a large number of tokens, e.g., 2655 for our dataset. Another possible representation is to encode a floorplan as a set of room polygons. However, we observed that predicting rooms as sequences of corners may lead to artificial room shapes as shown in Figure 1(a). We propose to employ a novel floorplan representation that slices the floorplan into a set of boxes by scanning the floorplan along the vertical axis (or y-axis). We denote this box set as $\mathcal{B} = \{b_i\}_{i=1}^{M}$, where $b_i = \{b_i^j\}_{j=1}^{N_m}$ and $N_m$ is the number of boxes $b_i^j$ in the $m^{th}$ room. Examples of the slicing boxes representation are depicted in Figure 3. The major advantage of this representation lies in its ability to maintain a manageable number of slicing boxes and query tokens ($\leq 50$ in our dataset). However, each floorplan may consist of a varying number of rooms, with each room containing a different number of slicing boxes. This poses a challenge in clustering the slicing boxes to their respective rooms. The unordered nature of the rooms, combined with the variable number of rooms and slicing boxes, makes it difficult to assign room IDs for the classification task. To tackle this challenge, we transform the original box classification problem into a 2D coordinate regression problem. Specifically, we employ one network to predict the room center for each slicing box, and another network to predict all the room centers from the density map. We compute the room center coordinate as the bounding box center for each room in the floorplan, represented as $\mathbb{C} = \{c_i\}_{i=1}^{M}$, where M denotes the number of rooms and $c_i \in [-1, 1] \times [-1, 1]$. This allows us to cluster the slicing boxes based on the predicted room centers.

## 3.3 Network architecture

Figure 4 illustrates the SLIBO-Net's architecture, comprising two primary components: the center transformer $\mathcal{T}_{center}$ and the box transformer $\mathcal{T}_{box}$. The center transformer $\mathcal{T}_{center}$ is responsible for predicting the coordinates of all room centers based on the density map, while the box transformer $\mathcal{T}_{box}$ predicts both the slicing boxes and their corresponding room center coordinates. Subsequently, a post-processing step is implemented to merge the estimated slicing boxes into a complete floorplan with distinct room layouts, utilizing the predicted room center points. Each component are elaborated in the next subsections.

**Center transformer.** First, we feed a $256 \times 256$ density map into a ResNet-50 feature extractor and generate one $16 \times 16$ feature map. Following the encoder-decoder transformer architecture, we feed the extracted feature maps into the transformer encoder block. The output tokens of the transformer encoder block are then used for cross attention in the transformer decoder block. In order to query all the room centers from the density map, we use $\bar{M}$ learnable embeddings as the input tokens of the transformer decoder. The output tokens of the transformer decoder are further processed using two auxiliary MLP heads. Specifically, we use: (i) *room center* MLP head $\mathcal{F}_{room}^{center}$ that outputs the coordinates of rooms' bounding box center $\mathbb{C}^{room} = \{c_i^{room}\}_{i=1}^{\bar{M}}$; and (ii) *room prob* MLP head $\mathcal{F}_{room}^{prob}$ that outputs the probabilities $\mathbb{P}^{room} = \{p_i^{room}\}_{i=1}^{\bar{M}}$ representing the probability of valid room center.

**Box transformer.** As for the box transformer $\mathcal{T}_{box}$, we follow the same architecture of center transformer $\mathcal{T}_{center}$ except the number of input tokens of the transformer decoder and the output MLP heads. We query all the slicing boxes from the density map via $\bar{N}$ learnable embeddings. All the output tokens of the box transformer $\mathcal{T}_{box}$ are further processed using two auxiliary MLP heads.

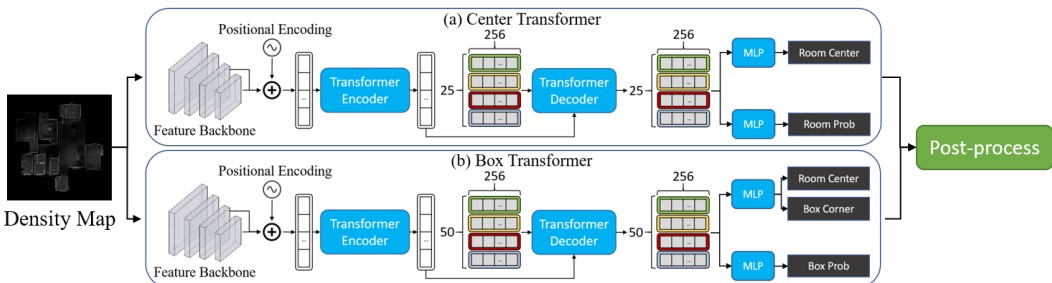

Figure 4: **Network architecture.** SLIBO-Net takes a 2D density map as input, and outputs a real-scale floorplan as the final result. (a) The center transformer $\mathcal{T}_{center}$ outputs a set of room centers and the probabilities for them to be valid. (b) The box transformer $\mathcal{T}_{box}$ predicts a set of slicing boxes, corresponding room centers, and their probabilities of being valid. Room centers and slicing boxes are further post-processed to form the final floorplan.

We use (i) *box coord* MLP head $\mathcal{F}_{box}^{coord}$ that outputs six scalars at once. The first two scalars represent the corresponding room's center coordinates $\mathbb{C}^{box} = \{c_i^{box}\}_{i=1}^{\bar{N}}$ and the remain four scalars represent the upper-left and lower-right corners of slicing boxes $\mathcal{B} = \{b_i\}_{i=1}^{\bar{N}}$; and (ii) *box prob* MLP head $\mathcal{F}_{box}^{prob}$ that outputs the probabilities $\mathbb{P}^{box} = \{p_i^{box}\}_{i=1}^{\bar{N}}$ representing the probability of valid slicing boxes.

**Bipartite matching.** Note that our neural network would always predict 25 room centers ($\mathbb{C}^{room}$) and 50 slicing boxes, which generally would be more than the ground-truth room centers $\overline{\mathbb{C}}^{room}$ and slicing boxes $\mathcal{B}$, respectively. Thus, following DETR [6], we conduct a set-based bipartite matching between predictions and targets in the center transformer $\mathcal{T}_{center}$ and the box transformer $\mathcal{T}_{box}$. Since we use the same strategy in both of the transformers, we explain this step using the center transformer $\mathcal{T}_{center}$ as an example. Our goal is to optimize a bipartite matching objective on a permutation function $\sigma(\cdot) : \mathbb{Z}_+ \to \mathbb{Z}_+$ which maps $\bar{M}$ prediction indices to $M$ targets:

$$\sigma^* = \arg\min_{\sigma} \sum_{i=1}^{\bar{M}} \mathbb{1}_{\{\sigma(i)\leq M\}}[\lambda_1 d_1(c_i^{room}, \bar{c}_{\sigma(i)}^{room}) + \lambda_2 d_2(p_i^{room})] \tag{1}$$

where $d_1(\cdot, \cdot)$ represents $L_1$ distance between predicted and target room center coordinates, $d_2()$ represents the focal loss between predicted and target room center probabilities, and $\mathbb{1}_{\{\cdot\}}$ is an indicator function. Finally, we use the Hungarian Matching algorithm to calculate the minimal cost $\sigma^*$ that maps $\bar{M}$ predicted room centers to $M$ target room centers and $\bar{N}$ predicted slicing boxes to $N$ target slicing boxes.

### 3.4 Post-processing

Here, we describe the post-processing steps in Figure 5 to reconstruct the floorplan using the slicing boxes $\mathcal{B}$ and room centers $\mathbb{C}^{room}$ obtained in the previous step by the SLIBO-Net.

The first step in the post-processing phase is to cluster the slicing boxes $\mathcal{B}$ into rooms. Given the predicted room centers $\mathbb{C}^{room}$ from the center transformer $\mathcal{T}_{center}$ and the predicted slicing boxes $\mathbb{C}^{box}$ corresponded rooms' centers $\mathbb{C}^{box}$ as the input, we can cluster the slicing boxes $\mathcal{B}$ into different rooms by assigning the $\mathbb{C}^{box}$ to the nearest room centers $\mathbb{C}^{room}$ from the center transformer $\mathcal{T}_{center}$.

After clustering the boxes into different rooms, we fill the gaps between the bottom and top of any two adjacent boxes in the same room with a box whose height is the vertical distance between them and whose width is the intersection length of them.

Finally, we union together the boxes for each room to form a room area $A_r$. We then overlay $A_r$ on the cell complex, which allows us to calculate the intersection ratio for each room per cell. By assigning each cell to the room with the largest intersection ratio, we can obtain a room representation by tracing the boundary formed by the cell groups. For non-connected components within a single room, we choose the component with the largest area. A cell remains empty if no room has more than 50% overlap.

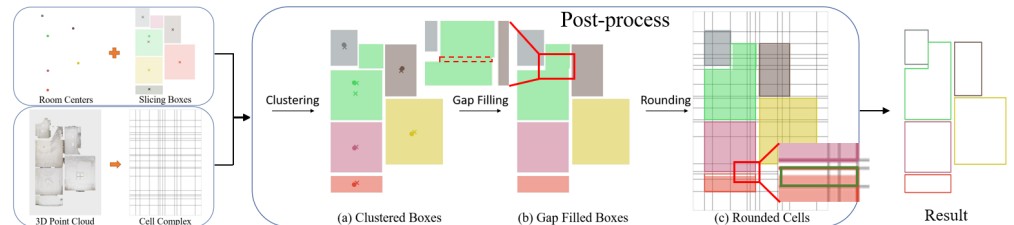

Figure 5: **Illustration of the post-processing.** We use cell complex and SLIBO-Net's output to get the floorplan. Same-color boxes are one room. (a) Crosses show slicing boxes' rooms' centers, dots show predicted room centers. (b) Box fills gap between boxes, red dashed box shows its size. (c) Colored boundaries are final result by boxes. Cells in green box belong to red room if they intersect more with red room and intersection ratio is over 50%.

## 3.5 Loss functions

We first elaborate on the loss functions we use for the center transformer $\mathcal{T}_{center}$ based on the optimal permutation $\sigma^*$ from the bipartite matching. For the center transformer $\mathcal{T}_{center}$, we use the index set $\{i; \sigma^*(i) \leq M\}$ representing the best matching prediction indices. For the coordinate regression, we use the smooth $L_1$ loss $\mathbb{L}_{smooth}(\cdot, \cdot)$ which is defined as follow:

$$\mathbb{L}_{smooth} = \begin{cases} \|X - Y\|_1 + 0.5(X - Y)^2, & \text{if } \|X - Y\|_1 < 1 \\ \|X - Y\|_1 + \|X - Y\|_1 - 0.5, & \text{otherwise} \end{cases} \tag{2}$$

where $X$ is the prediction and $Y$ is the label.

**Room center loss** calculates the smooth $L_1$ loss with ground truth room center $\overline{c}^{room}$:

$$\mathcal{L}_{rcenter}^{(i)} = \mathbb{1}_{\{\sigma^*(i) \leq M\}} \mathbb{L}_{smooth}\big(c_i^{room}, \overline{c}_{\sigma(i)}^{room}\big) \tag{3}$$

**Room prob loss** calculates the focal loss for room probability of being valid:

$$\mathcal{L}_{rprob}^{(i)} = -\mathbb{1}_{\{\sigma^*(i) \leq M\}}(1 - p_i^{room})^\gamma \log p_i^{room} - \mathbb{1}_{\{\sigma^*(i) > M\}}(p_i^{room})^\gamma \log(1 - p_i^{room}) \tag{4}$$

Thus, the final loss $\mathcal{L}_{rtotal}$ of the center transformer $\mathcal{T}_{center}$ is defined as below:

$$\mathcal{L}_{rtotal} = \sum_{i=1}^{M} \lambda_1 \mathcal{L}_{rcenter}^{(i)} + \lambda_2 \mathcal{L}_{rprob}^{(i)} \tag{5}$$

where $\lambda_1$ and $\lambda_2$ are the hyperparameters for weighting the loss functions.

Next, we describe the loss functions we compute for the box transformer $\mathcal{T}_{box}$ based on the optimal permutation $\sigma^*$ from the bipartite matching. For the box transformer $\mathcal{T}_{box}$, we use the index set $\{i; \sigma^*(i) \leq N\}$ representing the best matching prediction indices.

**Box room center loss** calculates the smooth $L_1$ loss for each slicing box room center with ground truth room center $\overline{c}^{box}$:

$$\mathcal{L}_{bcenter}^{(i)} = \mathbb{1}_{\{\sigma^*(i) \leq N\}} \mathbb{L}_{smooth}\big(c_i^{box}, \overline{c}_{\sigma(i)}^{box}\big) \tag{6}$$

**Box prob loss** calculates the focal loss for each slicing box probability of being valid:

$$\mathcal{L}_{bprob}^{(i)} = -\mathbb{1}_{\{\sigma^*(i) \leq N\}}(1 - p_i^{box})^\gamma \log p_i^{box} - \mathbb{1}_{\{\sigma^*(i) > N\}}(p_i^{box})^\gamma \log(1 - p_i^{box}) \tag{7}$$

**Box corner loss** calculates the smooth $L_1$ loss for all the corners of each slicing box:

$$\mathcal{L}_{bcorner}^{(i)} = \mathbb{1}_{\{\sigma^*(i) \leq N\}} \mathbb{L}_{smooth}\big(b_i^{corner}, \overline{b}_{\sigma(i)}^{corner}\big) \cdot w_{\sigma(i)} \tag{8}$$

where $b_i^{corner}$ represented all 4 corners of a slicing box. We assume a conversion from the original upper-left and lower-right representation of the slicing boxes $b_i$. We set $w_{\sigma(i)}$ to 1 if and only if the corner $\overline{b}_{\sigma(i)}^{corner}$ equals any corner in the floorplan; otherwise, it is 0.

Table 1: **Quantitative comparison of floorplan reconstruction on Structured3D [25] dataset.** The best results are in bold font. * means using cell complex to round the floorplan.

| Method | Room↑ | | | Corner↑ | | | Angle↑ | | | Delta↓ | | |
|---|---|---|---|---|---|---|---|---|---|---|---|---|
| | Prec. | Rec. | F1 | Prec. | Rec. | F1 | Prec. | Rec. | F1 | Prec. | Rec. | F1 |
| Floor-SP [8] | 0.89 | 0.88 | 0.885 | 0.81 | 0.73 | 0.768 | 0.80 | 0.72 | 0.758 | **0.01** | 0.01 | **0.01** |
| MonteFloor [22] | 0.956 | 0.944 | 0.950 | 0.885 | 0.772 | 0.825 | 0.863 | 0.754 | 0.805 | 0.022 | 0.018 | 0.02 |
| HEAT [9] | 0.969 | 0.940 | 0.954 | 0.817 | 0.832 | 0.824 | 0.776 | 0.790 | 0.783 | 0.041 | 0.042 | 0.041 |
| RoomFormer [24] | 0.977 | 0.965 | 0.971 | **0.889** | **0.851** | **0.870** | 0.826 | 0.791 | 0.808 | 0.063 | 0.06 | 0.062 |
| RoomFormer* [24] | **0.991** | 0.976 | 0.983 | 0.851 | 0.826 | 0.838 | 0.840 | **0.816** | 0.828 | 0.011 | 0.01 | **0.01** |
| Ours | **0.991** | 0.978 | 0.984 | **0.889** | 0.821 | 0.854 | **0.878** | 0.812 | **0.844** | 0.011 | **0.009** | **0.01** |

Table 2: **Quantitative comparison of floorplan reconstruction on Structured3D dataset with real-scale measurements (0.1m).** The best results are in bold font.

| Method | Room↑ | | | Corner@0.1 ↑ | | | Angle@0.1 ↑ | | | Delta@0.1 ↓ | | |
|---|---|---|---|---|---|---|---|---|---|---|---|---|
| | Prec. | Rec. | F1 | Prec. | Rec. | F1 | Prec. | Rec. | F1 | Prec. | Rec. | F1 |
| HEAT [9] | 0.969 | 0.94 | 0.954 | 0.434 | 0.436 | 0.435 | 0.417 | 0.419 | 0.418 | 0.017 | 0.017 | 0.017 |
| RoomFormer [24] | 0.977 | 0.965 | 0.971 | 0.637 | **0.612** | 0.624 | 0.608 | 0.585 | 0.596 | 0.029 | 0.027 | 0.028 |
| Ours | **0.991** | **0.978** | **0.984** | **0.652** | 0.602 | **0.626** | **0.648** | **0.598** | **0.622** | **0.004** | **0.004** | **0.004** |

**Box IoU loss** calculates the GIoU loss [6] $\mathbb{L}_{giou}$ for all slicing boxs with ground truth slicing box $\bar{b}_i$:

$$\mathcal{L}_{biou}^{(i)} = \mathbb{1}_{\{\sigma^*(i) \leq N\}} \mathbb{L}_{giou}(b_i, \bar{b}_{\sigma(i)}) \tag{9}$$

Thus, the final loss $\mathcal{L}_{btotal}$ of the box transformer $\mathcal{T}_{box}$ is defined as below:

$$\mathcal{L}_{btotal} = \sum_{i=1}^{N} \lambda_3 \mathcal{L}_{bcenter}^{(i)} + \lambda_4 \mathcal{L}_{bprob}^{(i)} + \lambda_5 \mathcal{L}_{bcorner}^{(i)} + \lambda_6 \mathcal{L}_{biou}^{(i)} \tag{10}$$

where $\lambda_{\{3,\ldots,6\}}$ are the hyperparameters for weighting the loss functions.

## 4 Experiments

### 4.1 Experimental settings

**Dataset.** We conducted experiments on a large-scale indoor synthetic dataset called Structured3D [25]. Following the approach in [9, 24], we divided the complete dataset into 2991 training samples, 250 validation samples, and 241 test samples, taking into account the ignore list specified in the HEAT [9] codebase. By employing the pre-processing method described in Section 3.1, we obtained a density map of size $256 \times 256$ for each sample, with pixel values normalized to the range of $[0, 1]$.

**Baselines.** We compare our method with the following competing methods: **Floor-SP** [8] - scores taken from [24], **MonteFloor** [22] - scores taken from [24], **HEAT** [9] - scores based on their official pre-trained weights, **RoomFormer** [24] - scores based on their official pre-trained weights.

**Evaluation metrics.** We follow the Room/Corner/Angle evaluation method from MonteFloor and HEAT to measure the quality of floorplan reconstruction. However, the existing metrics suffer from a scale-variant nature as they calculate results based on a normalized $256 \times 256$ pixel space, without considering the actual size of each floorplan. Consequently, this leads to a more lenient evaluation for floorplans with more rooms or larger sizes. Therefore, we propose the use of real-scale measurements, such as meters, as the threshold for Corner/Angle evaluation. For instance, Corner@0.1 denotes the measurement with a threshold of 0.1 meter.

We found that the current corner evaluation solely focuses on the distance to the ground truth corner, disregarding the shape connectivity between corners. This approach can lead to scenarios where there are multiple "correct" corners but inadequate connectivity (Figure 1(a)), resulting in a low Angle score. Hence, we additionally introduce the Delta metric that measures the global structure of the

Table 3: **Ablation study on loss function of box transformer.** The best results are in bold font.

| Method | Room↑ | | | Corner↑ | | | Angle↑ | | |
|---|---|---|---|---|---|---|---|---|---|
| | Prec. | Rec. | F1 | Prec. | Rec. | F1 | Prec. | Rec. | F1 |
| w/o Box IoU loss | 0.981 | 0.964 | 0.972 | 0.756 | 0.774 | 0.765 | 0.740 | 0.758 | 0.749 |
| w/o corner mask | 0.988 | 0.975 | 0.981 | 0.882 | **0.821** | 0.850 | 0.871 | 0.811 | 0.84 |
| Ours | **0.991** | **0.978** | **0.984** | **0.889** | **0.821** | **0.854** | **0.878** | **0.812** | **0.844** |

Table 4: **Ablation study on post-processing.** The best results are in bold font.

| Method | Room↑ | | | Corner↑ | | | Angle↑ | | |
|---|---|---|---|---|---|---|---|---|---|
| | Prec. | Rec. | F1 | Prec. | Rec. | F1 | Prec. | Rec. | F1 |
| Baseline | 0.991 | 0.978 | 0.984 | 0.889 | 0.821 | 0.854 | 0.878 | 0.812 | 0.844 |
| cluster w/ GT room corner | **0.994** | **0.987** | **0.99** | **0.894** | **0.831** | **0.861** | **0.884** | **0.822** | **0.852** |
| w/o gap filling | **0.991** | 0.977 | **0.984** | 0.887 | 0.815 | 0.849 | 0.875 | 0.805 | 0.839 |
| w/o cell complex rounding | 0.97 | 0.956 | 0.963 | 0.745 | 0.745 | 0.745 | 0.72 | 0.72 | 0.72 |
| w/ gap filling + rounding | **0.991** | **0.978** | **0.984** | **0.889** | **0.821** | **0.854** | **0.878** | **0.812** | **0.844** |

floorplan by computing the discrepancy between Corner and Angle scores. Thus, a lower Delta score indicates a better shape for the resulting floorplans.

**Implementation details.** We implemented our model in PyTorch and trained our center transformer $\mathcal{T}_{center}$ on 2 NVIDIA V100s for 2 days. The box transformer $\mathcal{T}_{box}$ was trained on 8 NVIDIA V100s for 1.9 days. We use the Adam optimizer with $b_1$=0.9 and $b_2$=0.999. The learning rates of two transformers and the ResNet-50 are 2.5e-4, and 1e-5, and we train the center transformer for 5000 epochs, box transformer for 16000 epochs with batch size 128 and 123 for each GPU. We empirically set $\lambda_1 = 1, \lambda_2 = 1$ in Equation 5, $\lambda_3 = 1, \lambda_4 = 1, \lambda_5 = 5., \lambda_6 = 1$ in Equation 10, $\gamma = 2$ in Equation 4, and $\bar{M} = 25, \bar{N} = 50$ in Section 3.3.

### 4.2 Floorplan reconstruction performance

**Quantitative comparison.** As shown in Table 1, our SLIBO-Net outperforms all the baselines in the overall F1 score. Compared to RoomFormer, which has the best F1 score among all baselines, our model improves the F1 score of Room by $1.3\%$ and Angle by $3.6\%$, while maintaining a comparable precision of Corner. Since RoomFormer fails to generate good room shapes (i.e., low Angle score), we round its results by our cell complex, denoted as RoomFormer*. Compared to RoomFormer*, our model achieves better F1 scores in all metrics. It is worth highlighting that our model exhibits a substantial performance boost of $5.2\%$ in the Delta F1 score when compared to RoomFormer. This advancement indicates our model's ability to generate more plausible and realistic floorplans.

**Real-scale evaluation.** As shown in Table 2, our model attains the highest scores across most of the metrics, including Room, Corner, Angle, and Delta, with a particular emphasis on the Delta scores. This improvement can be attributed to the incorporation of cell complex regularization, which effectively enhances our model's capabilities under more stringent evaluation conditions.

**Qualitative comparison.** The qualitative results are shown in Figure 6. Compared to HEAT, our model can reconstruct the correct floorplan structure and preserve the wall thickness , while HEAT fails to reconstruct the walls. Compared to RoomFormer, our model preserves all structural details without any self-intersection, whereas RoomFormer generates some self-intersections that are semantically implausible. Please refer to the supplementary for additional comparisons.

### 4.3 Ablation Studies

**The effectiveness of loss functions.** In this experiment, we evaluate the necessity of two loss functions. First, we remove the Box IoU loss (Equation 9) in our total loss function (Equation 10) of the box transformer. Second, we eliminate the corner mask by setting $w_{\sigma(i)}$ to 1 in Equation 8.

Table 5: **Quantitative comparison and computational efficiency by varying number of tokens in box transformer.** The best results are in bold font.

| # tokens | Time (sec./epoch)↓ | | Room↑ | | | Corner↑ | | | Angle↑ | | |
|---|---|---|---|---|---|---|---|---|---|---|---|
| | Training | Inference | Prec. | Rec. | F1 | Prec. | Rec. | F1 | Prec. | Rec. | F1 |
| 50 | **38.36** | **6.923** | **0.991** | **0.978** | **0.984** | **0.889** | **0.821** | **0.854** | **0.878** | **0.812** | **0.844** |
| 100 | 42.84 | 7.243 | 0.974 | 0.959 | 0.967 | 0.846 | 0.808 | 0.827 | 0.831 | 0.792 | 0.811 |
| 200 | 50.92 | 7.620 | 0.966 | 0.950 | 0.958 | 0.827 | 0.782 | 0.805 | 0.805 | 0.775 | 0.790 |
| 300 | 59.97 | 7.850 | 0.962 | 0.950 | 0.956 | 0.812 | 0.765 | 0.789 | 0.793 | 0.769 | 0.781 |

Table 6: **Quantitative comparision on the mean-shift clustering method.** Rows 4 to 7 present mean-shift clustering results using various bandwidth parameters. The best results are in bold font.

| Method | Room↑ | | | Corner↑ | | | Angle↑ | | |
|---|---|---|---|---|---|---|---|---|---|
| | Prec. | Rec. | F1 | Prec. | Rec. | F1 | Prec. | Rec. | F1 |
| Ours | **0.991** | 0.978 | **0.984** | **0.889** | 0.821 | **0.854** | **0.878** | 0.812 | **0.844** |
| Bandwidth=1 | 0.979 | **0.986** | 0.982 | 0.881 | 0.821 | 0.850 | 0.870 | 0.811 | 0.840 |
| Bandwidth=1.2 | 0.982 | 0.983 | 0.983 | 0.883 | 0.821 | 0.851 | 0.872 | 0.811 | 0.841 |
| Bandwidth=1.25 | 0.986 | 0.982 | **0.984** | 0.887 | **0.823** | **0.854** | 0.876 | **0.813** | **0.844** |
| Bandwidth=1.3 | 0.986 | 0.979 | 0.982 | 0.885 | 0.819 | 0.851 | 0.874 | 0.810 | 0.841 |

As shown in Table 3, we obtain the best performance with the Box IoU loss and the corner mask. The Box IoU loss significantly boosts our performance, which agrees with previous work [20]. Our slicing box representation is compatible with this Box IoU loss, unlike the corner representation. We attribute this Box IoU loss as one of the key factor for surpassing other methods in score.

**The necessity of each step in post processing.** In this experiment, we examine two aspects of our floorplan reconstruction method. First, we investigate how the accuracy of room center estimation affects the performance. Second, we evaluate the impact of gap filling and cell complex rounding in our post-processing step in Section 3.4. The upper block of Table 4 shows that using the ground truth room center for clustering the slicing boxes slightly improves the performance. The lower block of Table 4 shows that applying gap filling and cell complex rounding achieves the best performance. We observe that cell complex rounding significantly enhances the performance, indicating that cell complex regularization is a crucial component in our post processing.

**The impact of the number of tokens.** In this experiment, we examine how the number of tokens in the box transformer influences the computational efficiency and the final accuracy, as shown in Table 5. We observe that both the training time and inference time increase significantly as we increase the number of tokens. On the other hand, the accuracy also drops as we use more tokens. The main reason is that more tokens require more training time and introduce more dummy tokens, which makes it harder to reach the same accuracy within the same training time. These results confirm the importance of using a reasonable number of tokens to represent the floorplan, as discussed in Section 1.

**The effectiveness of center transformer.** In this experiment, we evaluate the necessity of our center transformer. We compared our results with another method that directly employs mean-shift clustering of the room centers estimated by the box transformer. As shown in Table 6, the conventional clustering approach experiences numerical fluctuations because of the varying bandwidth parameter. Overall, our method surpasses the performance of the mean-shift clustering approach.

## 5 Conclusions

We propose SLIBO-Net, a transformer-based method for floorplan reconstruction from 3D point clouds. Our method uses a novel floorplan representation that slices the floorplan into boxes, which reduces the tokens and improves the room shape supervision. We further use cell complex derived from the 3D point cloud as regularization and post-process to enhance the local details. We advocate a

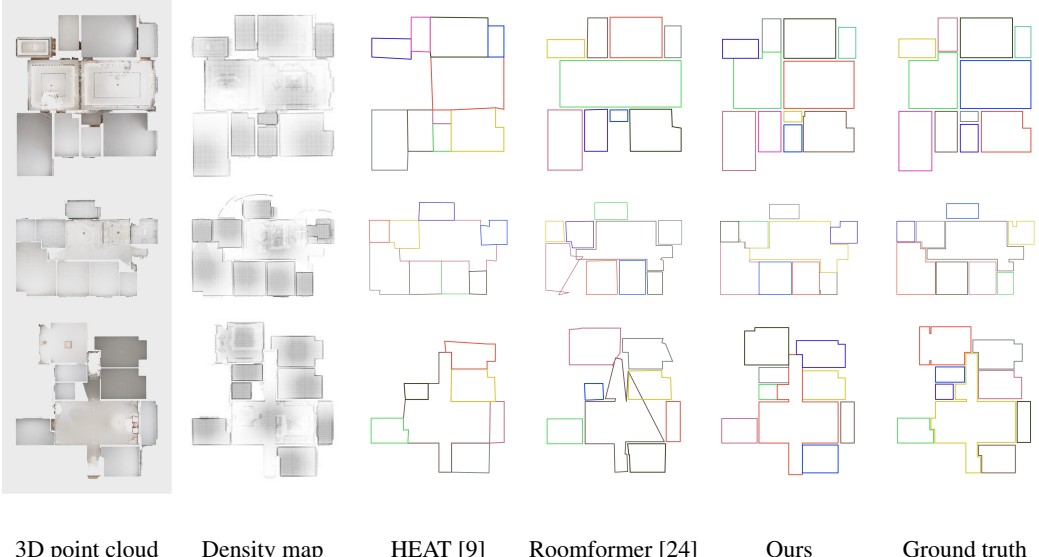

| 3D point cloud | Density map | HEAT [9] | Roomformer [24] | Ours | Ground truth |

Figure 6: **Qualitative evaluations on Structured3D.** Visual comparisons with competing methods.

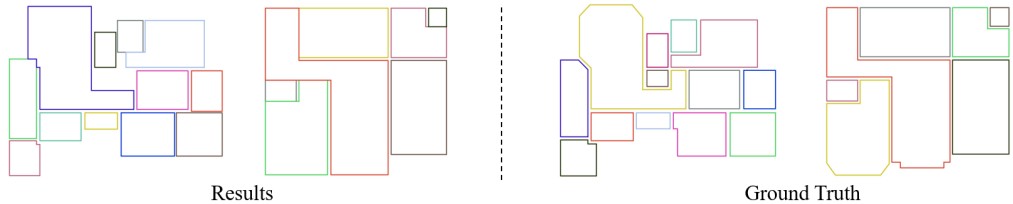

| Results | Ground Truth |

Figure 7: **Failure cases.** Left: our method struggles in reconstructing non-Manhattan layouts. Right: ground truth. One possible approach would be to extend the cell complex and post-processing to include non-Manhattan lines.

scale-independent metric that measures the local details better. Our method produces more meaningful and realistic floorplans and achieves state-of-the-art performance on various evaluation metrics.

**Limitations.** Our method is limited by its inability to handle non-Manhattan layouts (see Figure 7). Moreover, our network relies on a two-branch framework that constrains the reconstruction performance by the room center transformer.

**Future work.** Our method has 7 hyper-parameters in our loss functions that have not been thoroughly tuned. We intend to optimize these parameters to improve the performance. Additionally, we would like to integrate the cell complex into the box representation to avoid the need for the rounding step. However, this would require a major change in the data format for our current method, so it needs further investigation. Finally, we aim to make the post-processing more capable of handling non-Manhattan layouts.

**Broader impact.** This work significantly improves the efficiency and accuracy of floorplan reconstruction, which can benefit various applications in scene understanding, robotics, VR/AR, and construction industry. However, it also poses some risks, as the reconstruction is biased through training data and other factors.

# Acknowledgments and Disclosure of Funding

The project was funded in part by the National Science and Technology Council of Taiwan (110-2221-E-007-061-MY3, 110-2221-E-007-060-MY3, 112-2425-H-007-002-). We gratefully acknowledge Taiwan Web Service Corporation (TWSC) for their generous contribution of cloud GPU computing resources to support our research.

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
