# SLIBO-Net: Floorplan Reconstruction via Slicing Box Representation with Local Geometry Regularization Supplemental Material

**Jheng-Wei Su**[1], **Kuei-Yu Tung**[1], **Chi-Han Peng**[2], **Peter Wonka**[3], **Hung-Kuo Chu**[1,†]
[1]National Tsing Hua University [2]National Yang Ming Chiao Tung University
[3]King Abdullah University of Science and Technology
[†]Correspondence: hkchu@cs.nthu.edu.tw

## 1 Implementation details of competing methods

We compare our method with four competing methods in Table 1 of the main paper. Below, we provide more details about how we obtain the scores on Structured3D [6] for each method.

- **Floor-SP** [1] extracts geometry primitives from density maps using deep neural networks and optimizes the floorplan graph structure with room-wise coordinate descent. We use the evaluation score reported by [5].

- **MonteFloor** [3] applies MCTS to select room proposals that maximize an objective function combining the density map predicted by a deep network and regularization terms on the room shapes. We also use the score reported by [5].

- **HEAT** [2] is an end-to-end network that reconstructs structured floorplans from satellite or density images, using two shared-weights transformers that learn from image-level and geometric features separately. We evaluate their official pre-trained model and report the score.

- **RoomFormer** [5] is a transformer architecture that generates polygons of multiple rooms from a density map in parallel, without hand-crafted intermediate stages. We also evaluate their official pre-trained model and report the score.

## 2 Comparison on a real-world dataset.

Unlike synthetic datasets, real-world depth scanning often produces noisy and artifact-ridden meshes, which lead to projected density maps with more irregular sparsity patterns in many regions. In addition, real-world architectures have walls with varying thicknesses, which increases the difficulty for floorplan reconstruction. To address this challenge, we introduce a novel real-world dataset, **Gibsonlayout**, derived from the Gibson Environment Dataset [4]. It contains 1440 floors, of which we exclude 493 outdoor scenes or construction sites and use the remaining 947 ones. We label the layouts by starting from the panorama in each room and then aggregating all room layouts using the ground truth camera poses from the original Gibson Environment Dataset. As a result, we were able to obtain the floorplan for each floor. We split the full dataset into 599 training samples, 93 validation samples, and 255 test samples. The **Gibsonlayout** will be available online[1].

**Implementation details of competing methods.**

- **HEAT** [2]. In order to compare with HEAT [2] on the Gibsonlayout dataset, we followed the same procedure used on the Structured3D [6] dataset to obtain the density and normal

---

[1]https://ericsujw.github.io/SLIBO-Net/

37th Conference on Neural Information Processing Systems (NeurIPS 2023).

Table 1: **Quantitative comparison of floorplan reconstruction on Gibsonlayout dataset.** The best results are in bold font.

| Method | Room↑ | | | Corner↑ | | | Angle↑ | | | Delta↓ | | |
|---|---|---|---|---|---|---|---|---|---|---|---|---|
| | Prec. | Rec. | F1 | Prec. | Rec. | F1 | Prec. | Rec. | F1 | Prec. | Rec. | F1 |
| HEAT [2] | 0.755 | 0.711 | 0.733 | 0.626 | 0.584 | 0.605 | 0.522 | 0.491 | 0.506 | 0.105 | 0.094 | 0.099 |
| RoomFormer [5] | 0.717 | 0.601 | 0.654 | 0.584 | 0.415 | 0.485 | 0.414 | 0.288 | 0.339 | 0.17 | 0.127 | 0.145 |
| Ours | **0.857** | **0.837** | **0.847** | **0.638** | **0.614** | **0.626** | **0.607** | **0.582** | **0.594** | **0.032** | **0.032** | **0.032** |

Table 2: **Quantitative comparison of floorplan reconstruction on Gibsonlayout dataset with real-scale measurements (0.1m).** The best results are in bold font.

| Method | Room↑ | | | Corner@0.1 ↑ | | | Angle@0.1 ↑ | | | Delta@0.1 ↓ | | |
|---|---|---|---|---|---|---|---|---|---|---|---|---|
| | Prec. | Rec. | F1 | Prec. | Rec. | F1 | Prec. | Rec. | F1 | Prec. | Rec. | F1 |
| HEAT [2] | 0.755 | 0.711 | 0.733 | 0.31 | 0.293 | 0.301 | 0.266 | 0.252 | 0.259 | 0.044 | 0.041 | 0.042 |
| RoomFormer [5] | 0.717 | 0.601 | 0.654 | 0.282 | 0.199 | 0.233 | 0.229 | 0.156 | 0.186 | 0.053 | 0.043 | 0.048 |
| Ours | **0.857** | **0.837** | **0.847** | **0.411** | **0.393** | **0.401** | **0.401** | **0.383** | **0.392** | **0.01** | **0.009** | **0.009** |

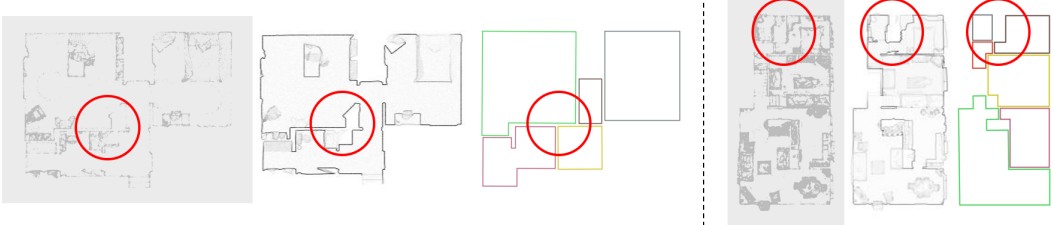

Figure 1: **Illustration of irregular boundaries.** The significant disparity between the irregular boundaries found in the meshes and density maps compared to the ground truth layouts can result in networks making implausible corner predictions. From left to right, we visualize 3D mesh, density map and ground truth layout.

maps. Additionally, we adjusted the floorplans into planar graphs using the pre-processing code provided by HEAT [2]. For training HEAT [2], we utilized their officially released code and default settings on the Structured3D [6] floorplans, with a batch size of 8.

- **RoomFormer** [5]. In order to conduct a comparison with RoomFormer [5], we deploy the same method used on the Structured3D [6] dataset to generate the density maps and ground truth. For training their official network, we utilize the default settings same as in Structured3D [6] with a batch size of 5. However, we make two adaptations specific to the Gibsonlayout dataset: the maximum polygon number is set to 40, and the maximum queries number is set to 2800. These adjustments account for the fact that the Gibsonlayout dataset has at most 35 rooms per floorplan and 68 corners per room.

**Quantitative comparisons.** As shown in Table 1, our SLIBO-Net outperforms all the baselines in all the metrics. Compared to HEAT [2], our model significantly improves the F1 score of Room by $11.4\%$, Corner by $2.1\%$, Angle by $8.8\%$, and Delta by $6.7\%$ on the pixel-level measurements. Moreover, our model achieves the best performance on the real-scale measurements as well, as shown in Table 2. Compared to HEAT [2], our model significantly improves the F1 score of Corner by $10\%$, Angle by $13.3\%$, and Delta by $3.3\%$ on the real-scale measurements.

Compared to the scores of Structured3D reported in the main paper, the scores of Gibsonlayout drop significantly across all the methods. We argue that this performance drop is caused by the real-world depth scanning, which could produce lots of artifacts and irregular boundaries in density maps (see Figure 1). These kinds of irregular boundaries are very different from the ground truth layouts, and thus pose a new reconstruction difficulty. We found that the corner-based methods, e.g., HEAT and RoomFormer, fail to reconstruct the correct floorplans and are easily affected by the irregular boundaries of density maps. However, our method is able to reconstruct reasonable layouts due to

our cell complex regularization and slicing box representation, which help us reconstruct floorplans without being affected by the irregular boundaries.

## 3 Visualization of floorplan reconstruction

We show more visual results and compare them with other competing methods on Structured3D (Figure 2) and Gibsonlayout (Figure 3). The results are presented in the following order: 3D point cloud or 3D mesh, density map, HEAT [2], RoomFormer [5], and our SLIBO-Net.

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

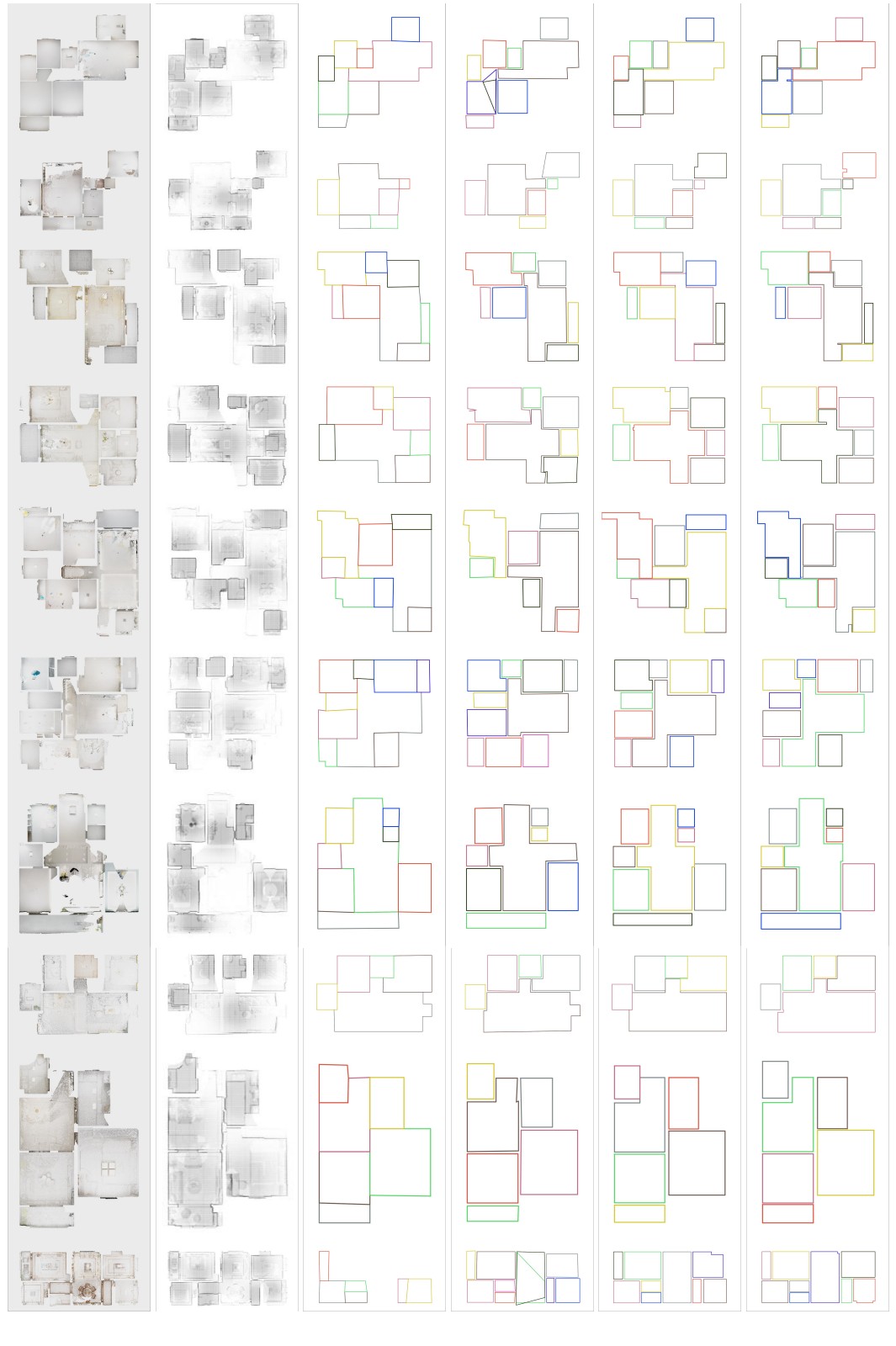

3D point cloud    Density map    HEAT [2]    RoomFormer [5]    Ours    Ground truth

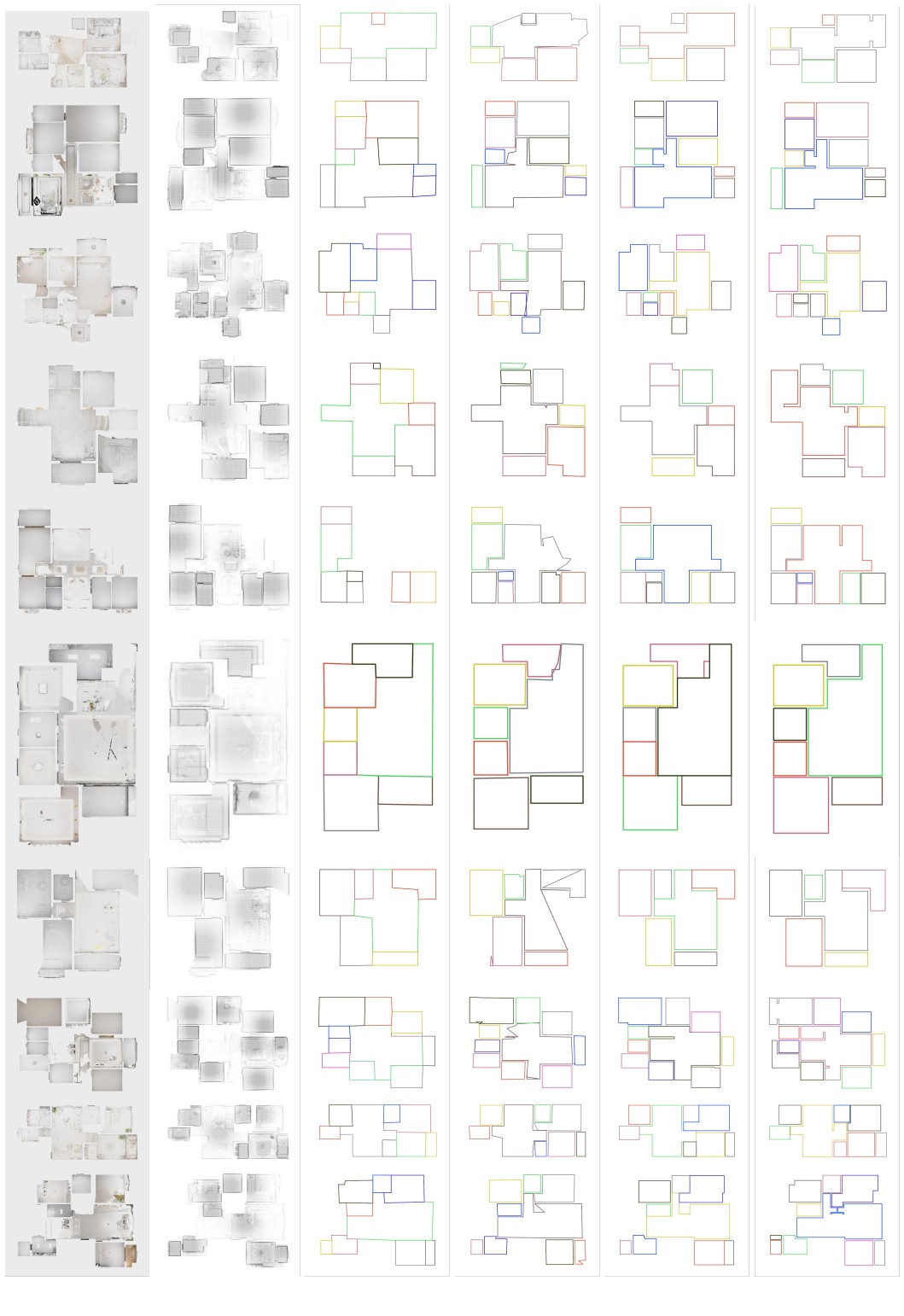

| 3D point cloud | Density map | HEAT [2] | RoomFormer [5] | Ours | Ground truth |

Figure 2: **Qualitative evaluations on Structured3D [6].** Visual comparisons with competing methods.

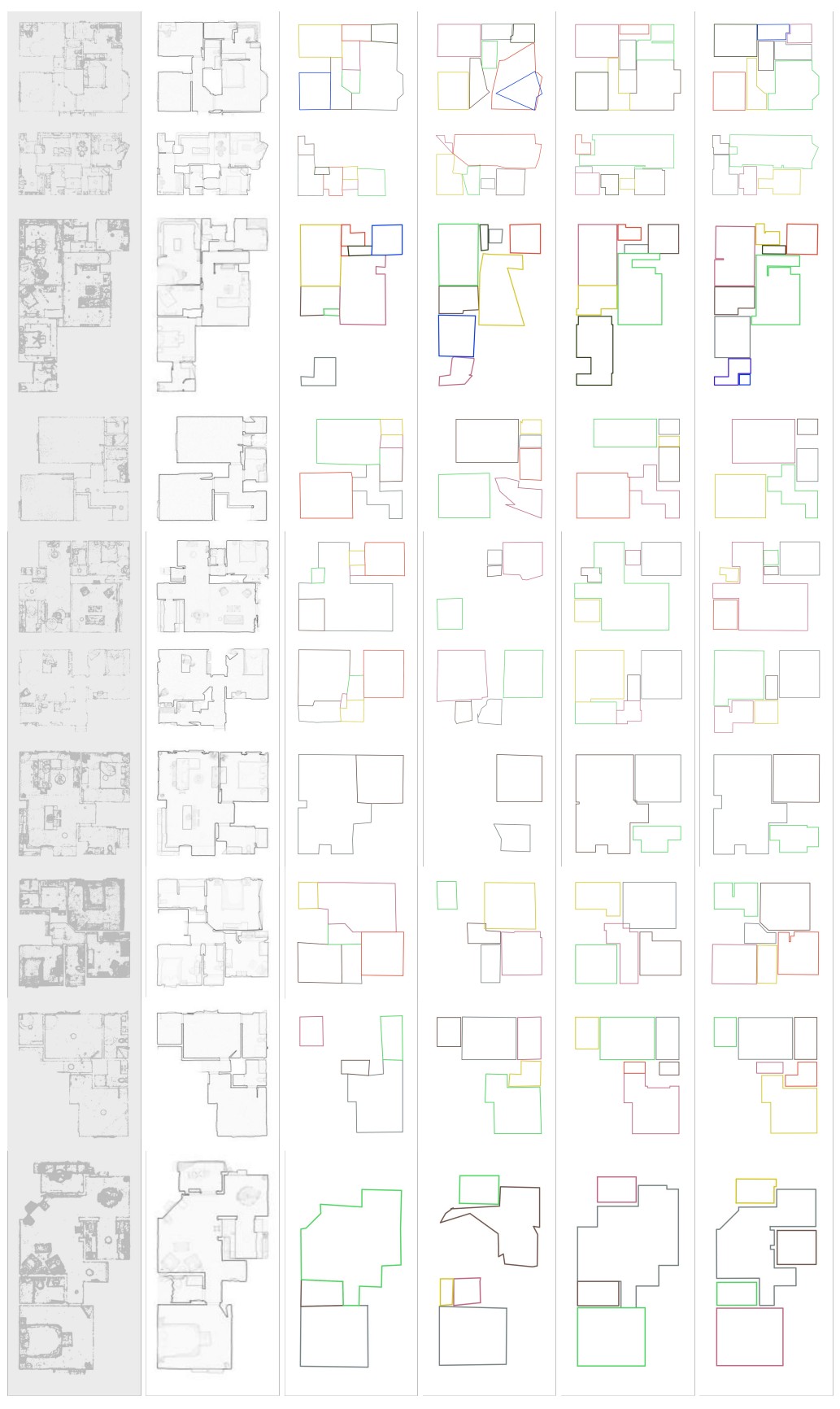

| 3D mesh | Density map | HEAT [2] | RoomFormer [5] | Ours | Ground truth |

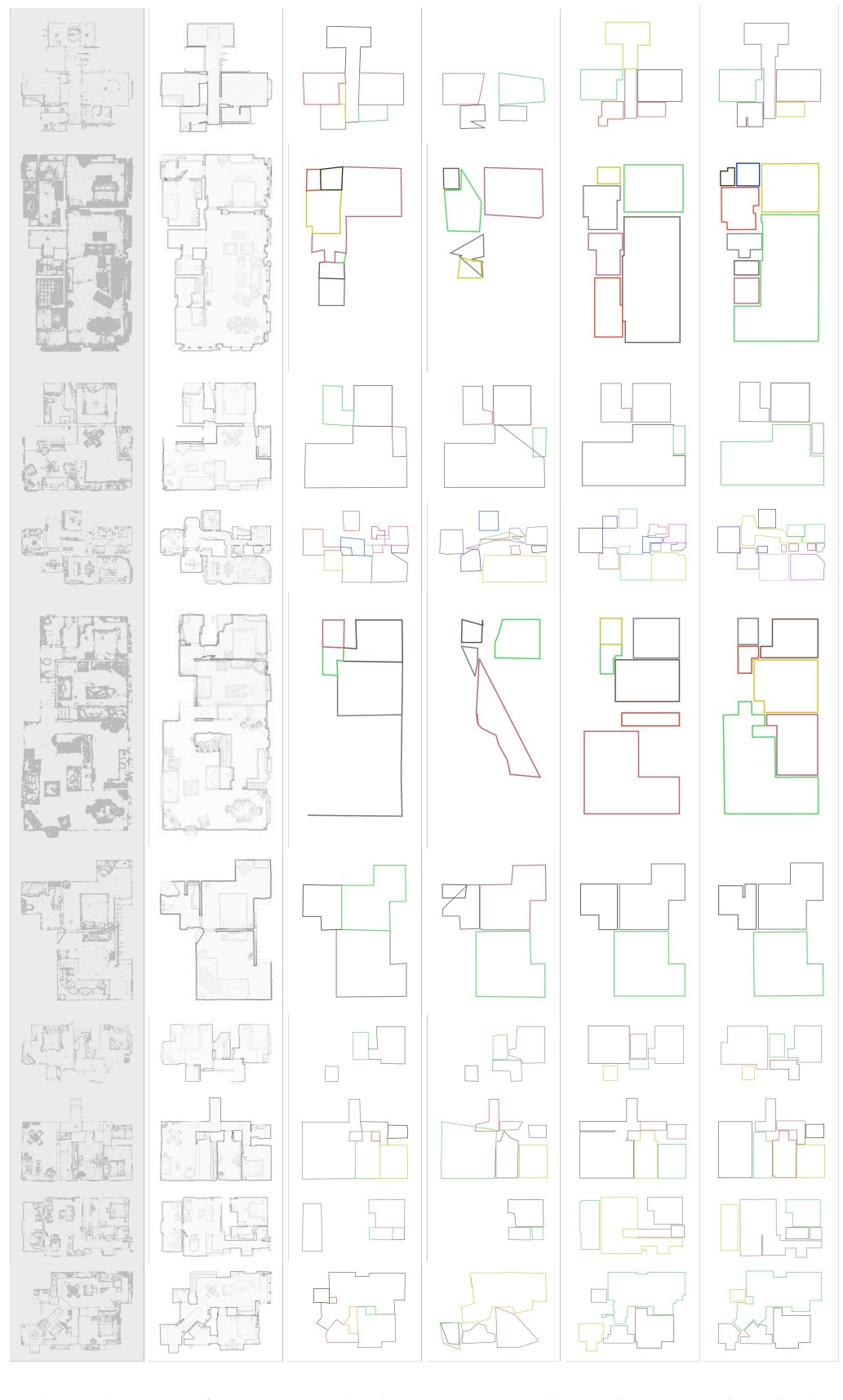

| 3D mesh | Density map | HEAT [2] | RoomFormer [5] | Ours | Ground truth |

Figure 3: **Qualitative evaluations on Gibsonlayout.** Visual comparisons with competing methods.