# OpenReview forum: "SLIBO-Net: Floorplan Reconstruction via Slicing Box Representation with Local Geometry Regularization"
_NeurIPS.cc/2023/Conference — NeurIPS 2023 poster_

### Official Review · Reviewer_qV7s · 2023-06-21

**Soundness:** 3 good
**Presentation:** 3 good
**Contribution:** 3 good
**Rating:** 7
**Confidence:** 5

**Summary:**

The paper presents a learning-based framework on floorplan reconstruction from point cloud. To tackle with the complex polygons emerge widely in floorplan, the polygon is decomposed to non-overlapping, vertical-slicing boxes as regression target of the networks. A bbox transformer and a room transformer are designed for bbox regression and room regression, respectively. A few post-processing steps are needed to intersect the two branches and integrate the bboxes into rooms. Extensive experiments are conducted on Structured3D dataset to verify its performance.

**Strengths:**

(1) The idea of dividing complex rooms into bboxes is novel and can effectively reduce the difficulty of learning the polygon corners from scratch.
(2) The overall pipeline is clear and straightforward, and well presented.
(3) The key challenges of the problem are well displayed in the introduction section, making it easier for readers to follow and future researchers to investigate on.

**Weaknesses:**

(1) Bbox assignment: the paper assigns the slicing bboxes to rooms by nearest distance search. However, this may not always be the case when some irregular polygons (the center living room shown in Figure 7 left). Some bboxes may be incorrectly matched with other rooms when there exists some neighboring rooms whose center is more adjacent. I wonder if there are more well-suited ways for bbox integration.
(2) Bbox corner regression: to represent a bbox, the paper learns to regress both the center and two corners (4 scalars). But there exists redundancy between their coordinates. Regressing height and width is enough if the bbox center is given. Please demonstrate the reason to regress both corners other than learning (h, w).
(3) A very recent work on floorplan generation (https://arxiv.org/abs/2207.13268) also employs the bbox (room) iou loss to enforce the correct structural regularity. Although it is a different problem, it is good to cite and discuss this paper.

**Questions:**

Overall I think the paper is novel and technically sound, with a clear pipeline for future research to follow. So currently I lean toward accepting this paper. However, I still expect authors’ feedback on my concerns listed in weaknesses. For question 1, I think a possible choice is to learn an assignment matrix during training between bboxes and rooms to encourage the matching process to be learned in a data-driven manner instead of only relying on center distances.

**Limitations:**

Authors have discussed their limitations on non-Manhattan floorplan reconstruction. It is indeed a challenging problem and may need some specified design to be proposed in the future.

---

> ### Author Rebuttal · Authors · 2023-08-09
>
> **W1**: *Bbox assignment: the paper assigns the slicing bboxes to rooms by nearest distance search. However, this may not always be the case when some irregular polygons (the center living room shown in Figure 7 left). Some bboxes may be incorrectly matched with other rooms when there exists some neighboring rooms whose center is more adjacent. I wonder if there are more well-suited ways for bbox integration.*
>
> **Reply**: Each slicing box predicted by the box transformer has two corners and a room center. The room center indicates the center of the room that the box belongs to, not the center of the box itself. Similarly, each room predicted by the center transformer has its own center. We match the boxes and rooms by finding the nearest distance between the room centers predicted by the box transformer and the room centers predicted by the center transformer.
>
> For example, for the red room on the right side of the Results in Figure 7 (We assume this is the “living room” mentioned in the comment), it should contain a room center predicted at the red dot (please refer to Figure 4 in the attached pdf). As for the green room, it should contain a room center predicted at the green dot. Although the blue box is closer to the room center of red room (red dot), the room center of blue box (blue cross) should be predicted closer to the room center of green room (green dot) than the room center of red room (red dot), so the blue box could be matched with the green room instead of the red room.
>
> **W2**: *Bbox corner regression: to represent a bbox, the paper learns to regress both the center and two corners (4 scalars). But there exists redundancy between their coordinates. Regressing height and width is enough if the bbox center is given. Please demonstrate the reason to regress both corners other than learning (h, w).*
>
> **Reply**: We use four corners of each slicing box as the input for our box corner loss, denoted as $b_i^{corner}$. As we discuss in the response to W1, each slicing box predicted by the box transformer has two corners and a corresponding room center. The raw output of each slicing box, denoted as $b_i$, represents the upper-left and lower-right corners of the slicing box. This means that we convert the format of the predicted slicing box before we compute the box corner loss, as shown in equation 8 of the main paper. Moreover, we multiply the loss with a mask term $w_{\sigma (i)}$ to ensure that we only compute the loss on the predicted corners whose corresponding target corners are the corners in the ground truth layout.
>
> **W3**: *A very recent work on floorplan generation [1] also employs the bbox (room) iou loss to enforce the correct structural regularity. Although it is a different problem, it is good to cite and discuss this paper.*
>
> **Reply**: Thank you for providing us with the related recent work. We will discuss and cite this paper in the revision.
>
> **Q1**: *Overall I think the paper is novel and technically sound, with a clear pipeline for future research to follow. So currently I lean toward accepting this paper. However, I still expect authors’ feedback on my concerns listed in weaknesses. For question 1, I think a possible choice is to learn an assignment matrix during training between bboxes and rooms to encourage the matching process to be learned in a data-driven manner instead of only relying on center distances.*
>
> **Reply**: There are different design choices to tackle the problem of clustering the boxes. The main challenge of learning an assignment matrix between the boxes and rooms lies in the unordered nature of the rooms, combined with the variable number of rooms and slicing boxes. This makes it difficult to assign room IDs during training, as mentioned in Section 3.2 of the main paper. As illustrated in the reply to W1, our method should be robust enough to address this issue.
>
> [1] End-to-end Graph-constrained Vectorized Floorplan Generation with Panoptic Refinement by Liu et al.

---

> > ### Comment · Reviewer_qV7s · 2023-08-18
> >
> > Thanks for the constructive feedback from authors.
> >
> > The feedback well resolved my concerns. For the question about the assignment matrix, instead of assigning IDs directly, I was referring to learning a similarity matrix between the recognized bboxes and rooms through the techniques such as contrastive learning. But anyway, the proposed design looks also feasible from my perspective.
> >
> > Based on the main paper and rebuttal, I would like to keep my original rating and accept this paper.

---

### Official Review · Reviewer_84jG · 2023-07-06

**Soundness:** 3 good
**Presentation:** 3 good
**Contribution:** 3 good
**Rating:** 6
**Confidence:** 5

**Summary:**

The paper introduces SLIBO-Net, a novel approach for reconstructing 2D floorplans from unstructured 3D point clouds. By employing a transformer-based architecture and an efficient floorplan representation, the method improves room shape supervision and manages token numbers effectively. The proposed approach achieves state-of-the-art results on the Structure3D dataset, generating floorplans with enhanced semantic plausibility and realistic reconstructions.

**Strengths:**

Technical Contribution: The authors introduce a novel approach for reconstructing 2D floorplans, tackling the challenges associated with semantic quality, efficient representation, and local geometric details. The proposed solutions are well-designed, showcasing a clear advancement over existing methods.

Presentation: The paper is well-structured and clearly presents the motivations, methodology, and experimental results. The technical concepts and terminologies are adequately defined and explained.

Evaluation: The authors conducted a thorough experimental evaluation, demonstrating the effectiveness of their proposed method. The comparisons with existing approaches and the state-of-the-art highlight the benefits of the proposed approach. The proposed scale-independent evaluation metric addresses a common limitation and ensures a fair assessment of performance across different floorplan sizes.

Contributions: The paper provides several notable contributions, including a novel transformer architecture for floorplan reconstruction, the integration of geometric priors for improved local geometric details, the adoption of a scale-independent evaluation metric, and the generation of semantically meaningful floorplans without self-intersections. These contributions offer valuable insights for future research.

Furthermore, the supplementary section of the paper introduces a valuable contribution in the form of a new labeled dataset called Gibsonlayout, which augments the Gibson dataset with annotated 2D floor-plans. The potential release of this dataset to the research community would be highly beneficial for advancing the field and promoting further research on the problem at hand. The availability of such a dataset would facilitate comparative evaluations, enable benchmarking, and encourage the development of more robust and accurate solutions.

**Weaknesses:**

The authors have effectively addressed a range of limitations in their paper, including the difficulties associated with handling non-Manhattan scenes and the seemingly arbitrary choice of the many hyperparameters. However, it is notable that the significant improvement in performance can be largely attributed to the Box IoU function, as acknowledged by the authors themselves. From a practical standpoint, this suggests that the additional technical contributions and associated complexity may have limited value for real-world applications.

**Questions:**

The description, design, and usage of the cell complex as presented in the paper may not be immediately intuitive or clear. It requires multiple readings to fully grasp how it is utilized in the final stages of the proposed method. To improve the clarity of this aspect, it would be advantageous for the authors to provide more explicit explanations and visual aids. For instance, incorporating a diagram that illustrates how and when the cell complex is employed would greatly enhance understanding and ensure a smoother comprehension of its role in the overall pipeline.

The authors' motivation in the proposed approach is heavily influenced by the transformer-based representation and the desire for a reasonable number of tokens. However, the impact of the selected number of tokens on the final performance, as well as the size and speed of the network, remains unclear. Conducting an ablation study that explores the relationship between the number of tokens and these factors would greatly enhance the effectiveness of the work. By systematically analyzing the effects of varying token counts, the authors can provide valuable insights into the trade-offs between representation size, computational efficiency, and model performance. Including such an ablation study would further strengthen the paper and provide a more comprehensive understanding of the proposed approach.

**Limitations:**

The authors have done good job discussing the most obvious limitations.

---

> ### Author Rebuttal · Authors · 2023-08-09
>
> **W1**: *It is notable that the significant improvement in performance can be largely attributed to the Box IoU function, as acknowledged by the authors themselves. From a practical standpoint, this suggests that the additional technical contributions and associated complexity may have limited value for real-world applications.*
>
> **Reply**: A possible direction for future work is to extend our box IoU function to a quad IoU function or a constrained quad IoU function, which could handle more challenging real-world non-Manhattan cases. This would allow our method to deal with more complex and irregular wall shapes, and improve the accuracy of our floorplan reconstruction. However, we would still consider our current solution very applicable to real-world applications.
>
> **Q1**: *The description, design, and usage of the cell complex as presented in the paper may not be immediately intuitive or clear. It requires multiple readings to fully grasp how it is utilized in the final stages of the proposed method. To improve the clarity of this aspect, it would be advantageous for the authors to provide more explicit explanations and visual aids. For instance, incorporating a diagram that illustrates how and when the cell complex is employed would greatly enhance understanding and ensure a smoother comprehension of its role in the overall pipeline.*
>
> **Reply**:  The cell complex plays a dual role in our method. First, it modifies the ground truth that we use for training, which derives the target slicing boxes. Second, it acts as a regularizer in the post-processing step, where we round the predicted slicing boxes using our cell complex. A complete system overview is illustrated in Figure 3 in the attached pdf.
>
> **Q2**: *The authors' motivation in the proposed approach is heavily influenced by the transformer-based representation and the desire for a reasonable number of tokens. However, the impact of the selected number of tokens on the final performance, as well as the size and speed of the network, remains unclear. Conducting an ablation study that explores the relationship between the number of tokens and these factors would greatly enhance the effectiveness of the work. By systematically analyzing the effects of varying token counts, the authors can provide valuable insights into the trade-offs between representation size, computational efficiency, and model performance. Including such an ablation study would further strengthen the paper and provide a more comprehensive understanding of the proposed approach.*
>
> **Reply**: The choice of prediction representation determines the maximum number of tokens, as it is not feasible to vary the number of tokens dynamically during the training of the transformer. The main challenge we saw was integrating the cell grid into the pipeline. Predicting a token per cell would already require 2655 tokens and predicting tokens per edge or corner would require even more. It requires 600 tokens for shared corners and edges (HEAT [1]), 800 tokens for non-shared corners (Roomformer [2]), and 75 tokens for slicing boxes and room centers (ours). To better demonstrate the benefits of using fewer tokens, we provide a comparison between different numbers of tokens and analyze their training and inference time of our box transformer in Table 6. The results show that we can achieve the best computational efficiency with the least tokens. Due to the time limitation, we have not conducted a comparison of the model performance. We will add this comparison in the revision.
>
> **Table 6** : **Computational efficiency of training time by varying number of tokens.**
>
> |Number of tokens|Training time|Inference time|
> |-|-|-|
> |50|38.36s / epoch|6.923s / epoch|
> |100|42.84s / epoch|7.243s / epoch|
> |200|50.92s / epoch|7.620s / epoch|
> |300|59.97s / epoch|7.850s / epoch|
> |1000|290.26s / epoch| 8.172s / epoch|
> |2655|1063.99s / epoch| 10.07s / epoch|
>
> [1] HEAT: Holistic Edge Attention Transformer for Structured Reconstruction by Chen et al.\
> [2] Connecting the Dots: Floorplan Reconstruction Using Two-Level Queries by Yue et al.

---

> > ### Comment · Reviewer_84jG · 2023-08-21
> >
> > I appreciate the effective rebuttal, I will keep my rating of weak accept.

---

### Official Review · Reviewer_EcBt · 2023-07-07

**Soundness:** 3 good
**Presentation:** 3 good
**Contribution:** 2 fair
**Rating:** 5
**Confidence:** 4

**Summary:**

This paper focuses on 2D floorplan reconstruction from 3D point clouds. The authors represent 2D floorplans as sliced ​​boxes and apply a transformer network to predict box characteristics, as well as post-processing steps to build the final floorplan. The authors conduct experiments on Structure3D and show better performance in both quantitative and qualitative terms compared to state-of-the-art methods.

**Strengths:**

- I like the idea of ​​introducing geometric priors (slicing boxes) into the design of neural network architectures. In my opinion, a combination of learning methods and pure geometric reasoning should be a better way to solve 3D related tasks. I'm happy to see better performance compared to other state-of-the-art method.

- In addition to the evaluation on synthetic Structure3D, the authors also introduce a novel real-world dataset (Gibsonlayout) in the supplement to further demonstrate the effectiveness of the proposed method compared to other state-of-the-art methods.

- The proposed method is technically sound. The authors also conduct ablation studies to verify the effectiveness of the design choices.

**Weaknesses:**

- I still don't understand why two parallel transformers are used instead of a single transformer for room center and slice box prediction. Perhaps it is better to show quantitatively that joint prediction is not a viable solution. Otherwise I find the proposed method a bit clumsy.

- It also better to conduct evaluation of inference time compared to other methods. I'm wondering how long the pre- and post-processing will take and the cost of running two transformers. If it takes significantly longer than others, it may not be desirable given the amount of improvement compared to other methods on Structure3D.

- Maybe an unnecessary requirement, it's also better to experiment on SceneCAD as other methods too. Although I know the authors propose a new dataset in the supplement, it should stand alone in the main text. Given the many heuristics introduced in pre- and post-processing, I want to make sure this is not tuned specifically for Structure3D and the proposed dataset.

- Minor issue, page 7, line 211, missing citation after Structure3D.

**Questions:**

Please see *Weaknesses

**Limitations:**

I found the proposed method a bit clumsy, with two parallel transformers instead of a single one. Inference time was not evaluated compared to other methods, so it is not known how the preprocessing and postprocessing steps affect the latency of this method.

---

> ### Author Rebuttal · Authors · 2023-08-09
>
> **W1**: *I still don't understand why two parallel transformers are used instead of a single transformer for room center and slice box prediction. Perhaps it is better to show quantitatively that joint prediction is not a viable solution. Otherwise I find the proposed method a bit clumsy.*
>
> **Reply**: Our method employs two transformers: the center transformer and the box transformer. The center transformer predicts a room corner for each input query, while the box transformer forecasts the boxes and their associated room centers. To consolidate these two transformers into a single branch, one could segment the input query into two subsets, directing each to a distinct transformer. The first subset would yield results akin to the center transformer, and the second subset to the box transformer. Nonetheless, this approach requires the transformer to handle diverse queries concurrently (room-wise tokens versus box-wise tokens), leading to inferior results compared to the original dual-transformer design, as evidenced in Table 3.
>
> **Table 3** : **Quantitative comparison on the single-transformer architecture.** The best results are emphasized in bold.
>
> |Method||Room$\uparrow$|||Corner$\uparrow$|||Angle$\uparrow$||
> |-|-|-|-|-|-|-|-|-|-|
> ||Prec.|Rec.|F1|Prec.|Rec.|F1|Prec.|Rec.|F1|
> |Our method|**0.991**|**0.978**|**0.984**|**0.889**|**0.821**|**0.854**|**0.878**|**0.812**|**0.844**|
> |Single transformer|0.970|0.966|0.968|0.853|0.796|0.823|0.840|0.784|0.811|
>
> Another possibility to build a single-transformer architecture is to directly employ a clustering algorithm on the room centers deduced from the box transformer. However, as discussed in the response to **Q3** of reviewer **mWNJ**, this alternative yields slightly inferior outcomes. As a result, we adopted the two-stage approach as our primary method.
>
> **W2**: *It also better to conduct evaluation of inference time compared to other methods. I'm wondering how long the pre- and post-processing will take and the cost of running two transformers. If it takes significantly longer than others, it may not be desirable given the amount of improvement compared to other methods on Structure3D.*
>
> **Reply**: We report the pre- and post-processing runtime in Table 4 and 5. In general, the computation time of all methods including ours is still very short compared to the time it takes to scan a complete floorplan with its corresponding processing. We conduct our experiments on a machine with an Intel i7-9700K @ 3.60GHz CPU, NVIDIA V100 GPU and 16 GB RAM. We also show the inference time for the two transformers in Table 4. Our method uses native transformers instead of deformable DETR [2], which makes our inference time slightly slower than other fully-neural methods, but still comparable. Overall, we believe that our method is worth the extra time, as it achieves significant improvement over other methods in terms of plausibility and accuracy.
> As we mainly focused on quality, we did not spend any time optimizing the pre-processing time as it is quite short overall.
>
> **Table 4** : **Inference time analysis on Structured3D dataset.** The unit of the time is Second.
>
> |Method|Pre-processing|Inference|Post-processing|Total|
> |-|-|-|-|-|
> |HEAT|1.87|0.11|-|1.98|
> |RoomFormer|0.57|0.01|-|0.59|
> |Ours|7.55|0.06|0.11|7.72|
>
> **Table 5** : **Timing analysis on pre-processing.** The unit of the time is Second.
>
> |Steps|Density Map|Clustering|Plane Fitting|Line Grouping|Non-axis-aligned Removal|Cell Complex|Total|
> |-|-|-|-|-|-|-|-|
> |Timing|0.57|3.21|0.36|3.22|0.08|0.11|7.55|
>
> **W3**: *Maybe an unnecessary requirement, it's also better to experiment on SceneCAD as other methods too. Although I know the authors propose a new dataset in the supplement, it should stand alone in the main text. Given the many heuristics introduced in pre- and post-processing, I want to make sure this is not tuned specifically for Structure3D and the proposed dataset.*
>
> **Reply**: We decided not to conduct experiments on the SceneCAD dataset due to the following 3 reasons. 1) Poor layout annotation quality. As depicted in Figure 2 of the attached pdf, a substantial number of layouts in the SceneCAD dataset are flawed and do not align with the density maps. These inconsistencies render the dataset noisy and less trustworthy. 2) Not suitable for the problem. SceneCAD offers only one room per floorplan and not complete floorplans. 3) Small size. SceneCAD only has 127 rooms. Gibsonlayout has 7,563 rooms with superior layout annotations. These annotations were derived from the panoramas in the Gibson Environment dataset, details of which are available in the supplementary material.
>
> **W4**: *Minor issue, page 7, line 211, missing citation after Structure3D.*
>
> **Reply**: Thanks for the reminder. We will add the missing citation in the revision.
>
> [1] Semantic Instance Segmentation with a Discriminative Loss Function by Brabandere et al.\
> [2] Deformable DETR: Deformable Transformers for End-to-End Object Detection by Zhu et al.

---

> > ### Comment · Reviewer_EcBt · 2023-08-19
> > **Re:**
> >
> > I appreciate the author's detailed response. I'm leaning towards acceptance.

---

### Official Review · Reviewer_mWNJ · 2023-07-10

**Soundness:** 3 good
**Presentation:** 3 good
**Contribution:** 3 good
**Rating:** 6
**Confidence:** 3

**Summary:**

The authors propose a new method for floorplan reconstruction from point clouds. Specifically, the method formulates the problem from predicting polygons for each room to predicting sliced boxes and their corresponding room centers by imposing the Manhattan grid assumption on the floorplans. The authors also use the prior information--cell complex, derived by extending the extracted wall segments from the input point clouds, to improve the results. The qualitative and quantitative results demonstrate that the proposed method outperforms the prior work by a significant margin.

**Strengths:**

1. The proposed method effectively improves the results using the Manhattan assumption and the prior knowledge, cell complex, obtained from the given point clouds. It is also a clever approach to use predicted room centers to cluster the boxes instead of predicting complex, non-rectangular polygons.

2. The figures are clear and helpful in understanding the methods and details.

3. The results outperform prior works by a significant margin, especially in the qualitative results' geometric details, such as wall gaps.

**Weaknesses:**

1. As the authors mentioned, the main limitation of this method is the Manhattan formulation.

2. Extending the local planes across the whole space might be unnecessary to form the cell complex. The extended planes will intersect at many empty locations. For example, the second line from the bottom in Figure 2(e) creates extra cell units on the bottom right, but it is unrelated to the room on the bottom right. Aren't the extracted wall segments already useful for post-processing? Or have the author tried using local grids (e.g., a plane is extended until it intersects with another plane) instead of global ones for the cell complex?

**Questions:**

1. It is unclear how the cell complex is used for regularization during training. None of the branches have a regularization term, and cell complex seems to be used only in post-processing.

2. The seems to be some redundancy in representing the box losses: why is computing loss on four corners required? Aren't top-left and bottom-right corners enough (i.e., xmin, ymin, xmax, ymax)?

3. It would be interesting to know how important the center transformer is. For example, the room assignment can be done using clustering algorithms on the predicted room centers.

4. How are the extracted wall segments for the cell complex aligned with the ground truth room shapes? Do the misalignments affect the accuracy?

5. The visual information in 3D is potentially helpful. Several works have reconstructed room shapes from panorama views, such as [1]. How are these approaches compared to the proposed method? For example, can one reconstruct the plan room-by-room instead of the whole floorplan?

[1] PSMNet: Position-aware Stereo Merging Network for Room Layout Estimation by Wang et al.

**Limitations:**

Yes, the authors have addressed the limitations.

---

> ### Author Rebuttal · Authors · 2023-08-09
>
> **W2**: *Extending the local planes across the whole space might be unnecessary to form the cell complex. The extended planes will intersect at many empty locations.*
>
> **Reply**: Extending local planes across the whole space has several advantages. Firstly, such unlimited expansion can propagate global geometric knowledge across different areas of the scene, capturing details not present in the initial data. Secondly, it can bridge the gaps in areas sparse in structural details, enhancing the accuracy of floorplan reconstructions (refer to Figure 1 in the attached pdf). Thirdly, this method aids in reconstructing extensive floorplans and multiple buildings by leveraging shared geometric insights. In addition, implementing limited local plane extension is not straightfoward due to the complex issue of plane intersection priorities.
>
> **Q1**: *It is unclear how the cell complex is used for regularization during training. None of the branches have a regularization term, and cell complex seems to be used only in post-processing.*
>
> **Reply**: The cell complex serves as a key modifier of the ground truth used in training and thus directly influences the objective function. We adopt the strategy outlined in Section 3.4 of the main paper, adjusting our ground truth by rounding to the cell complex prior to training. Consequently, this actively prompts our predictions to align with the cell complex. This integral use of the cell complex is pivotal in enhancing both the accuracy and robustness of our floorplan reconstruction technique.
>
> **Q2**: *The seems to be some redundancy in representing the box losses: why is computing loss on four corners required? Aren't top-left and bottom-right corners enough (i.e., xmin, ymin, xmax, ymax)?*
>
> **Reply**: Rather than calculating the box corner loss based on all four corners, we determine our box corner loss solely on those corners of the box that correspond to the ground truth layout's corners. This correspondence is denoted by the $w_{\sigma (i)}$ mask term in equation 8 of the main paper. Such an approach guarantees that our box corner loss exclusively fine-tunes the layout corners.
>
> **Q3**: *It would be interesting to know how important the center transformer is. For example, the room assignment can be done using clustering algorithms on the predicted room centers.*
>
> **Reply**: To highlight the importance of the center transformer, we compared our results with another method that directly employs mean-shift clustering to the room centers estimated by the box transformer. As shown in Table 1, the conventional clustering approach experiences numerical fluctuations because of the varying bandwidth parameter. Overall, our method surpasses the performance of the mean-shift clustering approach.
>
> **Table 1** : **Quantitative comparision on the mean-shift clustering method.** Rows 4 to 7 present mean-shift clustering results using various bandwidth parameters. The best results are emphasized in bold.
>
> |Method||Room$\uparrow$|||Corner$\uparrow$|||Angle$\uparrow$||
> |-|-|-|-|-|-|-|-|-|-|
> ||Prec.|Rec.|F1|Prec.|Rec.|F1|Prec.|Rec.|F1|
> |Our method|**0.991**|0.978|**0.984**|**0.889**|0.821|**0.854**|**0.878**|0.812|**0.844**|
> |Bandwidth=1|0.979|**0.986**|0.982|0.881|0.821|0.850|0.870|0.811|0.840|
> |Bandwidth=1.2|0.982|0.983|0.983|0.883|0.821|0.851|0.872|0.811|0.841|
> |Bandwidth=1.25|0.986|0.982|**0.984**|0.887|**0.823**|**0.854**|0.876|**0.813**|**0.844**|
> |Bandwidth=1.3|0.986|0.979|0.982|0.885|0.819|0.851|0.874|0.810|0.841|
>
> **Q4**: *How are the extracted wall segments for the cell complex aligned with the ground truth room shapes? Do the misalignments affect the accuracy?*
>
> **Reply**: In many cases, the extracted wall segments align with the ground truth. Nonetheless, some discrepancies between these segments and the ground truth do exist, influencing the "rounded ground truth" discussed in the response to Q1. The differences between the ground truth and its rounded counterpart is reported in Table 2. Notably, the accuracy of the rounded ground truth surpasses that of all existing methods. As a result, our predictions, which learns from the rounded ground truth, still outperforms other approaches.
>
> **Table 2** : **Quantitative comparison on the rounded ground truth.** The best results are emphasized in bold.
>
> |Method||Room$\uparrow$|||Corner$\uparrow$|||Angle$\uparrow$|||Delta$\downarrow$||
> |-|-|-|-|-|-|-|-|-|-|-|-|-|
> ||Prec.|Rec.|F1|Prec.|Rec.|F1|Prec.|Rec.|F1|Prec.|Rec.|F1|
> |RoomFormer|0.977|0.965|0.971|0.889|0.851|0.870|0.826|0.791|0.808|0.063|0.06|0.062|
> |Ours|0.991|0.978|0.984|0.889|0.821|0.854|0.878|0.812|0.844|0.011|0.009|0.01|
> |Rounded GT|**0.998**|**0.993**|**0.995**|**0.965**|**0.905**|**0.934**|**0.957**|**0.899**|**0.927**|**0.008**|**0.006**|**0.007**|
>
> **Q5**: *The visual information in 3D is potentially helpful. Several works have reconstructed room shapes from panorama views, such as [1]. How are these approaches compared to the proposed method? For example, can one reconstruct the plan room-by-room instead of the whole floorplan?*
>
> **Reply**: Reconstruction methods based on panorama images are generally investigated in a separate branch of work, e.g. [1, 2]. One of the main challenges of these methods is to register different rooms into a coherent floorplan. The registration process is often more difficult than the reconstruction of individual rooms, due to the lack of reliable features and correspondences across rooms. Therefore, the current bottleneck of registration limits the applicability of these methods to constitute full floorplans.
>
> [1] PSMNet: Position-aware Stereo Merging Network for Room Layout Estimation by Wang et al.\
> [2] GPR-Net: Multi-View Layout Estimation via a Geometry-Aware Panorama Registration Network by Su et al.

---

> > ### Comment · Reviewer_mWNJ · 2023-08-16
> >
> > Thank the authors for the response. Most of my concerns are addressed, and I will remain my rating as weak accept.

---

### Author Rebuttal · Authors · 2023-08-09

We are very grateful to all the reviewers for their valuable and insightful comments. To address all the questions, we have prepared a pdf document that contains figures to better illustrate our responses. Due to the format limitation of this pdf document, we will also explain all the figures below:

**Figure 1** (replying to the question **W2** of reviewer **mWNJ**): We show the point cloud and the corresponding cell complex of one case. The point cloud reveals that some of the occluded areas have very few points, which prevents the conversion of the local geometries to wall segments and the recovery of the correct layout structure by other methods. Our unlimited extended cell complex can propagate the global structure knowledge from the areas with dense point clouds to those occluded areas and provide more geometrical guidance for the subsequent steps of our pipeline.

**Figure 2** (replying to the question **W3** of reviewer **EcBt**): We present the flaws in the SceneCAD annotations, where the structures that are Manhattan are often marked as non-Manhattan. The middle column is the ground truth annotation for SceneCAD dataset, and the right column is the ground truth that we believe it should be. This kind of negligent labeling is very widespread on SceneCAD, lowering the quality of evaluation on SceneCAD.

**Figure 3** (replying to the question **Q1** of reviewer **84jG**): We demonstrate our system pipeline to clarify how the cell complex is employed in our method. The cell complex acts as a regularization mechanism during the training phase and is also utilized in the post-processing step to enhance the accuracy of the reconstructed floorplans.

**Figure 4** (replying to the question **W1** of reviewer **qV7s**): This figure illustrates the post-processing clustering process. Boxes are grouped into the same room based on a two-step assessment:

1. Their predicted room centers, as determined by the box transformer.
2. Proximity to the room centers estimated by the center transformer.

This indirect association method helps resolve situations where two boxes may be spatially near each other but do not actually belong to the same room.

---

### Decision · Program_Chairs · 2023-09-21

**Decision:**

Accept (poster)

**Comment:**

All the reviewers are initially positive about the paper. Rebuttals addressed most of the concerns and all the reviewers keep the original ratings. The area chair agrees with the positive assessment of the paper and recommends acceptance.